# Placental mRNA Expression of Neurokinin B Is Increased in PCOS Pregnancies with Female Offspring

**DOI:** 10.3390/biomedicines12020334

**Published:** 2024-02-01

**Authors:** Georgios K. Markantes, Evangelia Panagodimou, Vasiliki Koika, Irene Mamali, Apostolos Kaponis, George Adonakis, Neoklis A. Georgopoulos

**Affiliations:** 1Division of Endocrinology, Department of Internal Medicine, School of Health Sciences, University of Patras, 26504 Patras, Greece; vkoika@upatras.gr (V.K.); irenemamali@gmail.com (I.M.); or neogeo@upatras.gr (N.A.G.); 2Department of Obstetrics and Gynecology, School of Health Sciences, University of Patras, 26504 Patras, Greece; evakipanagod@hotmail.com (E.P.); akaponis@upatras.gr (A.K.); adonakis@upatras.gr (G.A.)

**Keywords:** polycystic ovary syndrome (PCOS), placenta, pregnancy, neurokinin B (NKB), kisspeptin (KISS1), placental expression

## Abstract

Current research suggests that polycystic ovary syndrome (PCOS) might originate in utero and implicates the placenta in its pathogenesis. Kisspeptin (KISS1) and neurokinin B (NKB) are produced by the placenta in high amounts, and they have been implicated in several pregnancy complications associated with placental dysfunction. However, their placental expression has not been studied in PCOS. We isolated mRNA after delivery from the placentae of 31 PCOS and 37 control women with term, uncomplicated, singleton pregnancies. The expression of KISS1, NKB, and neurokinin receptors 1, 2, and 3 was analyzed with real-time polymerase chain reaction, using β-actin as the reference gene. Maternal serum and umbilical cord levels of total testosterone, sex hormone-binding globulin (SHBG), free androgen index (FAI), androstenedione, dehydroepiandrosterone sulfate (DHEAS), Anti-Mullerian hormone (AMH), and estradiol were also assessed. *NKB* placental mRNA expression was higher in PCOS women versus controls in pregnancies with female offspring. *NKB* expression depended on fetal gender, being higher in pregnancies with male fetuses, regardless of PCOS. *NKB* was positively correlated with umbilical cord FAI and AMH, and *KISS1* was positively correlated with cord testosterone and FAI; there was also a strong positive correlation between *NKB* and *KISS1* expression. Women with PCOS had higher serum AMH and FAI and lower SHBG than controls. Our findings indicate that NKB might be involved in the PCOS-related placental dysfunction and warrant further investigation. Studies assessing the placental expression of *NKB* should take fetal gender into consideration.

## 1. Introduction

Polycystic ovary syndrome (PCOS) is the commonest endocrine abnormality in reproductive-aged women, affecting 5–18% of them, depending on the diagnostic criteria used [1,2]. The syndrome is characterized by considerable phenotypic heterogeneity: menstrual disturbances (mostly in the form of oligomenorrhea), the clinical manifestations of hyperandrogenemia (acne, hirsutism, androgenic alopecia), obesity, and/or insulin resistance (IR) that may affect the patients in different degrees [3]. Furthermore, women with PCOS frequently exhibit traditional risk factors for cardiovascular disease, including dyslipidemia, hypertension, metabolic syndrome, and type 2 diabetes [4]. PCOS is a major source of fertility problems, and it is the main cause of infertility related to anovulation [3]. Women with PCOS often need to use assisted reproduction technologies (ART) to achieve pregnancy and, even when this is accomplished, pregnancy is associated with an increased risk of adverse outcomes such as gestational diabetes, hypertension and preeclampsia (PE), miscarriage, and preterm delivery [5]. Anxiety, depression, and eating disorders are among the psychological comorbidities of the syndrome [6].

It is evident that PCOS is a highly prevalent disease, causing a significant burden to both patients and health care systems; hence, research aiming to understand its pathogenesis is expansive. Even though PCOS etiology is a topic attracting considerable attention, the cause of this disorder remains elusive. Several hypotheses pertaining to PCOS origins have been formulated, and most of them assume an interplay between genetic, epigenetic, and environmental factors [7]. Genome Wide Association Studies (GWAS) have identified several genetic loci with genetic susceptibility for PCOS, but the evidence supporting the relationship between these loci and the clinical manifestations of the syndrome are considered inconclusive [7]. Furthermore, it is estimated that these loci can explain less than 10% of PCOS heritability [8]. Obesity and exposure to endocrine-disrupting chemicals are also considered significant environmental contributors favoring PCOS development in genetically susceptible individuals [9]. Epigenetic mechanisms could also explain why only a subset of genetically predisposed individuals will manifest the syndrome, as it is known that factors external to the DNA can affect gene expression. A plausible theory is that the in utero hyper-exposure of the female fetus to androgens influences the expression of several genes, particularly those regulating ovarian steroid production, insulin action, and GnRH pulsatility, leading to a “re-programming” of the reproductive axis which later in life manifests as PCOS [7].

Indeed, data from animal studies have shown that prenatal exposure to excess androgen results in reproductive as well as metabolic disturbances that are typical to PCOS [10]. In humans, data are very limited but suggest that the fetuses of PCOS women are exposed to increased androgen levels in utero and this has been correlated with PCOS traits in later life [10].

The placenta, being the main steroidogenic organ in pregnancy, as well as a theoretical barrier to the overexposure of the fetus to maternally derived androgens (via androgen aromatization), could be implicated in the aforementioned theory of PCOS pathogenesis. It has been shown that women with PCOS demonstrate alterations in placental structure and function, and there is increasing evidence that androgen excess and insulin resistance, two characteristics of PCOS, may negatively affect the placenta [11]. As the placenta is a mediator of both pregnancy complications and developmental programming, mechanisms known to be involved in pregnancy complications such as PE and gestational diabetes might also be implicated in PCOS pregnancies.

Kisspeptin (KISS1) and neurokinin-B (NKB) are neuropeptides acting synergistically at the hypothalamus and stimulating pulsatile GnRH release. Their role is fundamental in puberty initiation, and they are also involved in the disturbed hypothalamic–pituitary function characterizing PCOS [12,13]. KISS1 and NKB levels increase in pregnancy due to placental production, and these peptides along with neurokinin receptors have been implicated in the pathogenesis of several pregnancy complications (PE, gestational diabetes, intrauterine growth restriction (IUGR), and preterm delivery) [14,15]. However, studies regarding the expression of these peptides in the human placenta in PCOS are lacking.

The aim of our study was to compare the placental expression of *KISS1, NKB*, and neurokinin receptors in women with PCOS and healthy pregnant women, and to correlate data from gene expression with the maternal and cord blood sex steroid levels.

## 2. Materials and Methods

### 2.1. Participants

This was a prospective, single-center case–control study. Participants were recruited from the Department of Obstetrics and Gynecology of the University Hospital of Patras, Greece, from January 2020 to December 2022. Only women with term, uncomplicated, singleton pregnancies, who gave birth to healthy babies were included in the study. The presence of PCOS prior to pregnancy was ascertained according to the Rotterdam criteria, when at least two of the following were present, i.e., oligo-anovulation, clinical and/or biochemical hyperandrogenemia, and polycystic ovarian morphology, after the exclusion of other pathologies with similar presentation [16]. The control group consisted of healthy women without a prior history of menstrual irregularity, clinical or biochemical hyperandrogenemia, or polycystic ovarian morphology. Exclusion criteria were any major medical condition, drug or alcohol use during pregnancy, and the use of hormonal or anti-diabetic medication up to 3 months prior to conception. Overall, 68 women participated in the study, with 31 having PCOS and 37 controls. The study was approved by the University Hospital of Patras Ethics Committee, and all participants provided written informed consent.

### 2.2. Placental Tissue Sampling and Gene Expression Analysis

Placental samples were collected within 15 min of placental delivery. We took three full-depth samples from each participant, sized 1 × 1 cm, from different areas at the middle point of the placental radius, away from infarcts or damage. We removed the maternal decidua and chorionic plate tissues and divided each sample into 2–3 pieces of 0.5–1 cm^3^. The samples were rigorously washed with normal saline solution to remove blood and were then submerged in RNAlater (QIAGEN, Hilden, Germany) solution and stored at 4 °C for 24 h and then at −20 °C until analysis.

We studied the mRNA expression of kisspeptin (*KISS1*), neurokinin B (*NKB*), and neurokinin receptors 1, 2, and 3 (*NK1R, NK2R, NK3R*). Total RNA was isolated from placental samples using the commercially available RNeasy Lipid Tissue Mini kit (QIAGEN, Hilden, Germany). In brief, tissue samples are homogenized in QIAzol Lysis Reagent. After the addition of chloroform, the homogenate is separated into aqueous and organic phases. The aqueous phase is collected, and ethanol is added to provide appropriate binding conditions. The sample is then subjected to a silica membrane column including a 15 min DNAse I treatment to avoid any DNA presence. Finally, RNA is eluted in RNase-free water. We estimated RNA concentration and purity by measuring the optical absorption at 260 nm and calculating the ratio 260/280 nm, respectively. Complementary DNA (cDNA) synthesis was carried out with the Transcriptor First Strand cDNA Synthesis Kit (04379012001; Roche, Basel, Switzerland) with a mixture of anchored-oligo(dT)18 primer (designed to bind at the beginning of the poly(A) tail and guarantee full-length cDNA synthesis) and 1 μg of total RNA. Quantitative real-time polymerase chain reaction (qRT-PCR) was conducted in the LightCycler 2.0 Instrument (Roche, Basel, Switzerland), using 50 ng of template cDNA and FastStart Universal SYBR Green 100 Master (Roche Hellas, Athens, Greece). PCR primers can be provided upon request for the genes *ACTB* (NM_001101), *KISS1* (NM_002256), *NKB* (NM_001006667), *NK1R* (NM_001058), *NK2R* (NM_001057), and *NK3R* (NM_001059). Reactions were run in triplicates. We used melting curve analysis to confirm the quality of the PCR reactions and the specificity of the primers. Relative gene expression was evaluated with the ΔΔCt method, using β-actin (*ACTB*) as a reference gene because of its suitability for this experimental setting [17].

### 2.3. Hormone Measurements

At delivery, blood samples were collected from the mother (from a peripheral vein) and from the umbilical cord for hormonal determinations. Samples were immediately centrifuged (2500× *g* for 10 min), and the serum was collected and stored at −80 °C until analysis. Total testosterone, sex hormone-binding globulin (SHBG), androstenedione, dehydroepiandrosterone sulfate (DHEAS), Anti-Mullerian hormone (AMH), and estradiol were measured using electrochemiluminescence quantization in the Cobas e601 analyzer (Roche Diagnostics^®^, Mannheim, Germany). The method follows a competition principle, in which the serum is incubated with a biotinylated antibody against the measured hormone and a hormone conjugate labeled with a ruthenium complex. The biotinylated antibodies form immune complexes with both the hormone that exists in the serum and the ruthenium-labeled conjugate. After the addition of streptavidin-coated microparticles, the complexes become bound to the solid phase via the interaction of biotin and streptavidin. In the measuring cell, the microparticles are magnetically captured onto the surface of the electrode. Unbound substances are removed, and the application of a voltage to the electrode induces chemiluminescent emission, which is measured using a photomultiplier, and the hormone concentration is determined via a calibration curve. The samples were assayed in a single large batch. The intra- and inter-assay precision CV (%) values were 2.1–14.8% and 2.5%–18.1% for testosterone, 1.1–1.7% and 1.8–4.0% for SHBG, 1.8–3% and 3.7–4.6% for androstenedione, 1.5–3.2% and 2.2–2.7% for DHEAS, 0.9–1.7% and 2.7–3.5% for AMH, and 1.1–6.7% and 1.9–10.6% for estradiol. The free androgen index (FAI) was calculated according to the formula: FAI = testosterone (nmol/L) × 100/SHBG (nmol/L)

### 2.4. Statistical Analysis

Data were analyzed using IBM SPSS Statistics for Windows, version 27.0 (IBM Corp., Armonk, NY, USA). Variables were tested for normality with the Kolmogorov–Smirnov test. Categorical data are presented as numbers (percentages) and continuous data as means ± standard deviations (SDs) (normally distributed variables) or as medians (interquartile ranges (IQRs)) (non-normally distributed variables). Comparisons between the two study groups were conducted using the independent samples *t*-test for normally distributed continuous data and the Mann–Whitney U test for non-normally distributed continuous data, while the chi-squared test was used for comparisons concerning categorical variables. Correlations were estimated by Pearson or Spearman correlation tests, as appropriate. All tests were 2-tailed, and a *p*-value of less than 0.05 was considered significant.

## 3. Results

The study included 68 women (PCOS, *n* = 31; controls, *n* = 37) with a mean age of 31.81 ± 5.49 years. The demographic data and pregnancy characteristics of the two groups are shown in Table 1. There was no significant difference between PCOS and control women regarding age, BMI, the presence of gestational diabetes, pregnancy duration, the mode of delivery, and offspring gender, weight, or length.

Women with PCOS had higher serum FAI and AMH and lower SHBG levels than controls, while no significant difference was observed between the two groups regarding serum total testosterone, androstenedione, DHEAS, and estradiol (Table 2). The umbilical cord blood hormone levels were studied separately in male and female offspring, as there are considerable differences between genders. In both the male and female subgroups, the cord blood hormone levels did not differ significantly between PCOS and control women’s offspring (Table 2).

Maternal serum total testosterone levels were strongly positively correlated with androstenedione (Spearman’s r = 0.875, *p* < 0.001), DHEAS (r = 0.385, *p* < 0.001), and AMH (r = 0.407, *p* < 0.001). AMH was positively correlated with testosterone, as well as with estradiol (r = 0.287, *p* = 0.018) and androstenedione (r = 0.454, *p* = 0.002). Maternal SHBG was negatively correlated with the first visit BMI (r = −0.305, *p* = 0.013), and positively with estradiol (r = 0.371, *p* = 0.002).

Umbilical cord total testosterone was positively correlated with maternal serum testosterone (r = 0.249, *p* = 0.043), DHEAS (r = 0.428, *p* < 0.001), and androstenedione (r = 0.329, *p* = 0.029). Cord blood DHEAS and androstenedione were also positively correlated with maternal serum androgens, while cord blood estradiol was positively correlated with maternal estradiol, testosterone, DHEAS, and androstenedione.

We did not detect a statistically significant difference in the placental mRNA expression of the studied genes between PCOS and control women, although *NKB* and *KISS1* showed a trend for increased expression in PCOS (*p* = 0.16 for *NKB* and *p* = 0.12 for *KISS1*) (Table 3). Similarly, there was no difference in the placental gene expression between PCOS and control women in the male offspring subgroup. However, in the female offspring group, *NKB* expression was significantly increased in PCOS women versus controls (Median 2^−ΔCt^ for relative expression to *ACTB* was 0.0008 for PCOS and 0.0001 for Controls, *p* = 0.021) (Table 3, Figure 1). Furthermore, placental *NKB* expression was higher in women with male offspring versus those with female offspring, regardless of their PCOS status (*p* = 0.034).

In the correlations of placental gene expression with demographic and pregnancy characteristics, *NK3R* expression was positively correlated with the maternal BMI at delivery (Spearman’s r = 0.409, *p* = 0.005). There was no correlation between placental gene expression and maternal serum hormone levels. Placental *NKB* expression showed a positive correlation with umbilical cord FAI (r = 0.356, *p* = 0.021) and AMH (r = 0.336, *p* = 0.028) levels, while *KISS1* expression was positively correlated with cord estradiol (r = 0.324, *p* = 0.032), testosterone (r = 0.432, *p* = 0.003), and FAI (r = 0.415, *p* = 0.006). Lastly, *NKB* and *KISS1* expression were strongly and positively correlated (r = 0.496, *p* < 0.001), while the expression of each neurokinin receptor was positively correlated with the expression of the other two (*NK1R* with *NK2R* and *NK3R*, and so forth).

## 4. Discussion

The present study showed that the placental mRNA expression of *NKB* is increased in women with PCOS versus controls in pregnancies with female offspring. It also confirmed that pregnant women with PCOS have higher serum AMH and higher amounts of circulating free, bioactive androgens (higher FAI and lower SHBG) compared to healthy pregnant women. Umbilical cord blood hormones did not differ between the groups in neither male nor female offspring. Notably, our study included only women with term, uncomplicated, singleton pregnancies.

The presence of a genetic component in PCOS has been anticipated since the 1960s, with studies showing the familial segregation of the syndrome [18]. Since then, several reports have confirmed these findings, demonstrating a high prevalence of PCOS in the first-degree relatives of the probands [19], as well as a high concordance in twins [8]. However, family and twin studies have failed to provide a consistent pattern of inheritance, while genetic loci identified using GWAS are estimated to explain only a small part of PCOS heritability [7,19].

The Barker or thrifty phenotype hypothesis was introduced in the 1990s, after a study showed that IUGR and low birth weight were correlated with cardiovascular disease that the fetus will develop when it becomes middle-aged [20]. This hypothesis supports that the intrauterine environment can “program” the future health of the fetus and determine the risk of chronic diseases, by inducing epigenetic changes in the fetal DNA. In the context of PCOS, this theory proposes that a hyperandrogenic intrauterine environment has epigenetic consequences affecting the expression of genes that regulate the future endocrine and metabolic functions of the fetus (i.e., GnRH pulsatility, folliculogenesis, the ovarian production of sex steroids, insulin resistance). This theory for PCOS inheritance has been supported by animal studies showing that the administration of androgens during pregnancy leads to the manifestation of PCOS-like phenotypes in adult life in the female offspring; results have been consistent across a variety of species, such as rodents, sheep, and monkeys [21,22,23]. These studies have proven that a prenatal androgen excess produces reproductive defects, such as morphological changes/increased kisspeptin expression and reduced sex steroid feedback in the hypothalamus and an increased LH secretion, leading to increased ovarian androgen production, anovulation, and infertility [10,23,24,25,26,27]. It has also been shown that an in utero androgen excess is associated with metabolic perturbations in adult life, i.e., beta cell dysfunction, insulin resistance, hyperinsulinemia, increased adiposity, and obesity have been reported [28,29,30]. In humans, obviously, there are and will be no studies exploring the effect of exogenous androgen administration in pregnancy. Some studies have longitudinally followed-up the offspring of hyperandrogenic mothers with PCOS and showed that the daughters of PCOS women exhibit PCOS features, like an elevated testosterone, an increased LH, and an increased ovarian volume [31]. Furthermore, the daughters of mothers with congenital adrenal hyperplasia (CAH), another disease state of intrauterine hyperandrogenemia, frequently manifest a PCOS phenotype [32]. Although data regarding the longitudinal metabolic phenotyping of PCOS or CAH offspring is sparse, insulin resistance, dyslipidemia, and an increased hospitalization for metabolic disorders have been reported in this population [33,34,35].

The placenta is the source of oxygen and nutrients for the fetus, and also the major producer of the steroid hormones required for the maintenance of pregnancy. As the main determinant of the intrauterine hormonal and metabolic/nutritional milieu, it is consequently implicated in the hypothesis linking intrauterine hyperandrogenemia with PCOS inheritance. In support of this view, placental insufficiency leads to infants born small for gestational age (SGA), which has been shown to be a predisposing factor for PCOS [36]. There is a great body of evidence exposing the central role of the placenta in a variety of pregnancy complications (spontaneous abortion, PE, preterm labor, IUGR, SGA) [11]. PCOS pregnancies are characterized by increased rates of such adverse outcomes which are directly related to placental dysfunction [5]. Therefore, it is possible that there are common pathogenic mechanisms/placental alterations underlying both these complications and PCOS. Khan et al. showed, for example, that there is an overlap of five proteomic biomarkers between PCOS and PE [37]; however, there are several mediators of placental pathophysiology that are still not fully investigated in PCOS [11].

There is no doubt that the process of placentation can be problematic in PCOS, as evidenced by studies demonstrating an increased occurrence of structural abnormalities in the placenta of women suffering from the syndrome. Palomba et al. found that PCOS placentae had significantly lower weight, thickness, density, and volume and an increased frequency of lesions such as fibrosis compared to those of healthy pregnant women [38]. Another study showed that PCOS was characterized by a higher incidence of placental anomalies associated with an increased hypoxic state, such as chorioamnionitis, funisitis, villitis, vascular thrombosis, infarction, and villous immaturity [39]. Notably, in both of these studies, the increased rate of placental anomalies in PCOS was not accounted for by other pregnancy complications, as the former study included only uncomplicated pregnancies, while in the latter, the adjustment for pregnancy complications was performed. Finally, a recent study examining the placentas of women with PCOS who underwent in vitro fertilization revealed that PCOS was associated with important anatomic as well as vascular placental abnormalities [40].

Apart from anomalies in placental structure, it seems that placental steroidogenic function is also impaired in PCOS. Androgen levels rise during physiologic pregnancy, serving as estrogen precursors but also in the initiation of parturition. Normally, maternal and fetal over-exposure to androgen is hampered by a significant increase in SHBG production by the maternal liver and a high level of placental aromatase activity [10]. However, it is possible that in pregnant PCOS women, the supra-normal ovarian androgen production exceeds the capacity of these mechanisms, facilitating the hyperexposure of the fetus to maternal androgens. Indeed, increased 3b-hydroxysteroid dehydrogenase and decreased aromatase activities have been observed in placental tissue from PCOS women; this was correlated with increased maternal serum androgens [41]. Furthermore, PCOS pregnancies are characterized by elevated androgen levels in the maternal serum [42,43,44], as well as in the amniotic fluid [45] compared to controls; studies investigating umbilical cord blood androgen levels and their correlation with maternal androgens have yielded conflicting results [11,44]. In our study, pregnant women with PCOS had significantly higher serum FAI, hence, an increased concentration of free androgens relative to controls; the levels of total testosterone, androstenedione, and DHEAS did not differ between the groups. Maternal serum testosterone was positively correlated with the other maternal androgens, a finding that could reflect the coordinated increase in all steps of steroidogenesis characterizing the end of gestation [10]. The positive correlation between maternal AMH and testosterone has been reported before in pregnant and non-pregnant women [46,47]. Although a causal relationship between AMH and testosterone has not been established, it has been suggested that testosterone could affect follicular growth [46], and that AMH might induce gestational hyperandrogenemia [47]. Not surprisingly, maternal SHBG concentrations were positively correlated with maternal estradiol and negatively correlated with maternal BMI; it is well known that estradiol increases and obesity decreases the hepatic SHBG production. With regard to umbilical cord hormone levels, there was no significant difference between the groups, regardless of the offspring gender. However, cord blood androgens were positively correlated with maternal serum androgens, underlining the orchestrated steroidogenesis of the maternal–placental–fetal unit, or even implying that increased maternal androgen levels could favor placental androgen production (as shown in animals) [10]. Aside from an altered steroidogenesis, the PCOS status affects placental function in a more generalized way, as proteomic analysis showed the differential expression between PCOS and control women in 258 placental proteins [44].

Little is known concerning the mechanisms involved in the placental dysfunction complicating PCOS, and most data come from animal models. Both animal and human placentae express the androgen receptor and are therefore susceptible to androgen effects [48]. The exogenous administration of androgens during gestation in the animal models of PCOS leads to decreased placental weight and fetal growth restriction [49,50,51,52,53,54], implying a negative effect of hyperandrogenemia on placental function and efficiency. The altered regulation of placental nutrient transport (decreased free fatty acid and amino acid transport and increased signal transducer and activator of transcription 3 (STAT3) expression) has been shown in prenatally androgenized rodents and monkeys [49,54,55]. Increased placental STAT3 signaling has also been found in humans with PCOS [56]. Despite increased free serum androgens, our PCOS women did not differ from controls in terms of the offspring weight and length; however, our study included only uncomplicated pregnancies, and therefore, the cases of placental insufficiency, IUGR, or SGA were excluded.

Furthermore, androgen administration results in the increased placental expression of estrogen and androgen receptors in rodents [51,57], further enhancing the actions of steroid hormones. Gestational hyperandrogenism has also been linked with placental vasculopathy; treatment with testosterone led to decreased uterine artery blood flow, elevated vascular resistance, and the increased expression of hypoxia-responsive genes in rats [52,53], while in sheep, it led to increased placental mRNA and protein expression of vascular endothelial growth factor (VEGF) [50]. Insulin resistance is another cardinal characteristic of PCOS, which might adversely affect placentation. Insulin resistance impairs human trophoblast invasion in vitro, while insulin sensitizers promote appropriate trophoblast migration and invasion [58]. Hyperandrogenic and insulin-resistant rats show disrupted trophoblast invasion/differentiation, associated with placental mitochondrial dysfunction and the over-production of reactive oxygen species [59]. Insulin resistance predisposes a woman with PCOS to gestational diabetes mellitus (GDM), which commonly complicates PCOS pregnancies [5]. GDM is in turn associated with placental dysfunction, via several mechanisms such as reduced placental apoptosis, impaired vasculogenesis, increased ischemia, and villous immaturity [11]. A small number of women with GDM were included in our study (four PCOS women and four controls). These women were diagnosed by an oral glucose tolerance test conducted at the 24th–28th week of pregnancy, none received insulin treatment, and all achieved excellent glycemic control with diet only. This mild abnormality in glucose metabolism would be expected to have minor, if any, effects on the placenta. The placental expression of the studied genes did not differ between women with and without GDM.

Another potential factor implicated in PCOS heritability and acting, at least in part, at the level of the placenta is AMH. The administration of AMH to pregnant mice led to increased maternal GnRH activity, increased LH and testosterone, and decreased estradiol and progesterone; at the placenta level, AMH treatment induced a decrease in aromatase and 3b-hydroxysteroid dehydrogenase and an increase in LH receptor expression; the female offspring of AMH-treated mice demonstrated an altered GnRH activity and the phenotypic features of PCOS [47]. In humans, AMH levels are normally low in pregnancy, reflecting ovarian suppression. However, in PCOS, high AMH levels persist and even increase in pregnancy [47]. It is therefore possible that an increased AMH contributes to maternal hyperandrogenemia via GnRH/LH activation. Moreover, since the human placenta expresses the AMH receptor type 2, an increased maternal AMH might block placental aromatase and contribute to fetal androgen overexposure. Our study confirmed that PCOS is characterized by the non-suppression of AMH during pregnancy, since we found that serum AMH levels measured at delivery are significantly higher in PCOS women versus controls.

KISS1 and NKB are increased in pregnancy as a result of placental production. Although the role of kisspeptin in regulating trophoblast invasion and embryo implantation is established [15], little is known regarding the function of NKB in normal pregnancy [14]. Kisspeptins may have a role in the pathogenesis of GDM, as lower KISS1 plasma levels and higher placental protein expression of KISS1 and the kisspeptin-1 receptor have been found in women with GDM [60,61]. KISS1 levels are elevated in preeclamptic compared to healthy pregnant women, and its concentrations correlate with the severity of PE [62,63], while *KISS1* mRNA expression is increased in the preeclamptic placenta [64,65]. Furthermore, a decreased maternal KISS1 has been correlated with fetal growth restriction [66]. KISS1 might also exert antiapoptotic effects on the placenta: in hypothyroid pregnant rats, the administration of kisspeptin suppresses the apoptotic effect of hypothyroidism, by blocking the activation of the inflammasome NLRP3 pathway [67].

It is well established that PE is characterized by significantly increased maternal NKB levels and by increased placental *NKB* mRNA and protein expression; the placental expression of neurokinin receptors is also increased in PE [14,68]. The chronic infusion of high-dose NKB in rats induces hypertension [69], implying a possible etiologic role of NKB in PE development. The concentration of NKB is increased in the maternal plasma of IUGR pregnancies [70], while an increased placental mRNA expression of *NKB* has been shown in preterm labor [71].

Although KISS1 and NKB are implicated in PCOS pathogenesis (both at the level of the hypothalamus [12,13] and at the level of the ovary [72]) and in several pregnancy complications characterized by placental dysfunction, data concerning their placental expression in PCOS are lacking. To our knowledge, our study was the first to assess the expression of these peptides in PCOS placentae. We found that the mRNA expression of *NKB* was increased in the placenta of women with PCOS and female offspring compared to controls. This is a very interesting finding, as altered NKB expression might be involved in adverse pregnancy outcomes complicating PCOS but could also be implicated in the syndrome’s heritability. When all women were examined together regardless of the offspring gender, there was a trend for increased placental expression of *NKB* and *KISS1*, but this did not reach statistical significance. *NKB* expression was higher in women with male versus those with female offspring, regardless of their PCOS status. Furthermore, *NKB* demonstrated a positive correlation with umbilical cord FAI and AMH levels, while *KISS1* was positively correlated with cord testosterone and FAI. These findings demonstrate that the placental expression of *NKB* and *KISS1* either depends on fetal androgen and/or AMH levels, or it is regulated by factors that also influence fetal hormones. In support of the former possibility, studies in animals have shown that intrauterine hyperandrogenemia induces morphological changes in kisspeptin/neurokinin B/dynorphin neurons and increases kisspeptin-positive cells in the hypothalamus [24,25]; furthermore, the exogenous AMH administration during gestation alters the hypothalamic GnRH network [47] (in which NKB and KISS1 are major components). Maybe the fact that the percentage of male offspring was higher in our control group (57% vs. 45% in PCOS) attenuated the difference in *NKB* and *KISS1* expression between PCOS and control women in the whole group analysis. The notion that fetal sex might affect the placental adaptation to external insults is not new; testosterone administration in pregnant sheep leads to fetal growth restriction more frequently in female fetuses, underscoring the possibility of sex-specific placental alterations in PCOS pregnancies [73]. In our study, the placental expression of the neurokinin receptors 1, 2, and 3 appeared to be higher in pregnancies with male versus female offspring, although this did not reach statistical significance. This finding should be carefully interpreted, as the placental expression of neurokinin receptors expression was very low relative to the reference gene (*ACTB*). Finally, we observed a strong positive correlation in the placental expression of *NKB* and *KISS1*, compatible with data from studies in rat placental cells showing that NKB upregulates *KISS1* mRNA expression [74].

Our study does not lack limitations. We used the Rotterdam criteria to diagnose PCOS, which led to the inclusion of all PCOS phenotypes in our cohort, increasing the heterogeneity of this subgroup and possibly affecting the results; for example, since increased maternal androgens are a potential mediator of placental dysfunction, placental alterations might differ in hyperandrogenic versus normoandrogenic PCOS women. PCOS was retrospectively diagnosed in this study, increasing the risk of recall bias. Furthermore, our study cannot answer whether the observed increase in placental *NKB* expression contributes to PCOS-related placental dysfunction, or it is a compensatory mechanism. In support of the former, NKB suppresses several proteins involved in antioxidant defense and in the inhibition of intravascular coagulation in human cytotrophoblast cells [75], suggesting that NKB might be a crucial mediator in placental dysfunction.

In conclusion, our study showed that *NKB* placental mRNA expression is increased in women with PCOS versus controls in pregnancies with female offspring. *NKB* expression depends on fetal gender, and it is positively correlated with placental *KISS1* expression as well as with umbilical cord blood FAI and AMH levels. More studies are needed to clarify the potential role of NKB in the placental abnormalities characterizing PCOS.

## Figures and Tables

**Figure 1 biomedicines-12-00334-f001:**
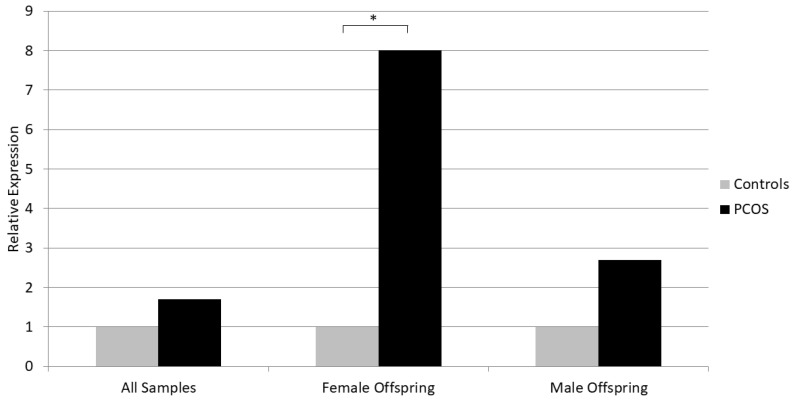
Relative placental mRNA expression of NKB in PCOS and control women. The expression level in controls is considered as reference. * *p* < 0.05 (Mann–Whitney U test).

**Table 1 biomedicines-12-00334-t001:** Demographic data and pregnancy characteristics of the PCOS and control women included in the study. VD: vaginal delivery; CS: cesarean section; M: male; and F: female. Categorical data are presented as number (percentage) and continuous data as mean ± SD.

	PCOS (*n* = 31)	Controls (*n* = 37)	*p* Value
Age (years)	31.52 ± 5.32	32.06 ± 5.78	0.677
BMI at first visit (kg/m^2^)	26.81 ± 5.06	25.27 ± 4.44	0.191
BMI at delivery (kg/m^2^)	31.98 ± 5.49	29.84 ± 4.44	0.105
Gestational diabetes	4 (12.9%)	4 (10.8%)	0.790
Delivery week	39 (2)	39 (2)	0.785
Mode of delivery (VD/CS)	15 (48.4%)/16 (51.6%)	19 (51.4%)/18 (48.6%)	0.808
Offspring gender (M/F)	14 (45.2%)/17 (54.8%)	21 (56.8%)/16 (43.2%)	0.274
Offspring weight (g)	3161.33 ± 555.14	3330.27 ± 462.90	0.179
Offspring length (cm)	49.75 ± 2.44	50.88 ± 2.27	0.139

**Table 2 biomedicines-12-00334-t002:** Maternal serum and umbilical cord blood hormone levels in PCOS and control women. SHBG: sex hormone-binding globulin; FAI: free androgen index; DHEAS: dehydroepiandrosterone sulfate; and AMH: Anti-Mullerian hormone. Data are presented as mean ± SD (normally distributed variables) or as median (IQR) (non-normally distributed variables).

Maternal Serum	PCOS (*n* = 31)	Controls (*n* = 37)	*p* Value
Total testosterone (ng/dL)	88.29 (98.77)	91.27 (70.99)	0.538
SHBG (nmol/L)	415.99 ± 135.61	478.20 ± 120.47	**0.049**
FAI	0.68 (0.40)	0.56 (0.67)	**0.048**
Androstenedione (ng/mL)	2.19 (2.10)	1.64 (2.12)	0.387
DHEAS (μg/dL)	105.97 ± 54.46	120.99 ± 72.82	0.347
AMH (pmol/L)	7.23 (5.41)	3.84 (6.07)	**0.012**
Estradiol (pg/mL)	8860 (16,301)	6698 (15,510)	0.310
Umbilical cord blood			
Female Offspring	PCOS (*n* = 17)	Controls (*n* = 16)	
Total testosterone (ng/dL)	142.54 ± 56.71	130.97 ± 37.27	0.501
SHBG (nmol/L)	32.16 (18.15)	32.50 (23.31)	0.624
FAI	14.57 (9.37)	12.69 (11.52)	0.468
Androstenedione (ng/mL)	0.52 ± 0.12	0.44 ± 0.11	0.185
DHEAS (μg/dL)	425.58 ± 194.81	353.89 ± 151.84	0.255
AMH (pmol/L)	1.50 (2.03)	1.19 (1.56)	0.624
Estradiol (pg/mL)	2577.19 ± 929.94	2959.82 ± 1095.36	0.295
Male Offspring	PCOS (*n* = 14)	Controls (*n* = 21)	
Total testosterone (ng/dL)	168.76 ± 69.34	166.62 ± 56.62	0.923
SHBG (nmol/L)	33.54 (9.02)	37.06 (12.43)	0.255
FAI	17.14 (9.13)	16.12 (6.73)	0.893
Androstenedione (ng/mL)	0.56 ± 0.24	0.48 ± 0.17	0.341
DHEAS (μg/dL)	394.01 ± 166.72	349.44 ± 161.11	0.449
AMH (pmol/L)	165.84 (80.60)	180.10 (90.90)	0.439
Estradiol (pg/mL)	3266.23 ± 1375.43	2730.19 ± 1769.12	0.363

Statistically significant *p* values are shown in bold.

**Table 3 biomedicines-12-00334-t003:** Placental mRNA expression of *NKB, NK1R, NK2R, NK3R*, and *KISS1* genes in PCOS and control women. Results are expressed as the median (IQR) relative expression to *ACTB* according to the ΔΔCt method.

All Samples	PCOS (*n* = 31)	Controls (*n* = 37)	*p* Value
*NKB*	0.0017 (0.04)	0.0010 (0.02)	0.160
*NK1R*	2.58 × 10^−5^ (5.3 × 10^−5^)	2.44 × 10^−5^ (9.0 × 10^−5^)	0.514
*NK2R*	3.66 × 10^−6^ (9 × 10^−6^)	4.14 × 10^−6^ (14 × 10^−6^)	0.298
*NK3R*	5.41 × 10^−5^ (18.8 × 10^−5^)	2.80 × 10^−5^ (7.4 × 10^−5^)	0.394
*KISS1*	0.0160 (0.07)	0.0079 (0.07)	0.120
Female Offspring	PCOS (*n* = 17)	Controls (*n* = 16)	
*NKB*	0.0008 (0.03)	0.0001 (0.0001)	**0.021**
*NK1R*	1.51 × 10^−5^ (2.9 × 10^−5^)	2.43 × 10^−5^ (7.0 × 10^−5^)	0.762
*NK2R*	0.77 × 10^−6^ (24 × 10^−6^)	1.57 × 10^−6^ (4 × 10^−6^)	0.579
*NK3R*	1.66 × 10^−5^ (37.0 × 10^−5^)	1.42 × 10^−5^ (6.3 × 10^−5^)	0.631
*KISS1*	0.0136 (0.03)	0.0022 (0.03)	0.315
Male Offspring	PCOS (*n* = 14)	Controls (*n* = 21)	
*NKB*	0.0135 (0.07)	0.0050 (0.02)	0.586
*NK1R*	5.62 × 10^−5^ (9.4 × 10^−5^)	3.56 × 10^−5^ (31.4 × 10^−5^)	0.867
*NK2R*	4.63 × 10^−6^ (12 × 10^−6^)	2.97 × 10^−6^ (5 × 10^−6^)	0.660
*NK3R*	8.40 × 10^−5^ (18.7 × 10^−5^)	2.20 × 10^−5^ (6.8 × 10^−5^)	0.363
*KISS1*	0.0435 (0.26)	0.0089 (0.08)	0.135

Statistically significant *p* values are shown in bold.

## Data Availability

Data are available upon request.

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
