# Peer review of "Placental mRNA Expression of Neurokinin B Is Increased in PCOS Pregnancies with Female Offspring"

_biomedicines, 2024, doi:10.3390/biomedicines12020334_

Round 1

Reviewer 1 Report

Comments and Suggestions for Authors

Comments about the manuscript:

“Placental mRNA Expression of Neurokinin B is Increased in PCOS Pregnancies with Female Offspring”

Polycystic ovary syndrome could occur in utero, and involve the placenta in particular through the production of kisspeptin and neurokinin B. The study presented here concerns a comparison of the expression of placental KISS1, NKB and neurokinin receptors in the placental mRNA of patients with polycystic ovarian syndrome and of healthy pregnant women, with the aim of correlating these gene expressions with maternal and cord blood sex steroid levels. The findings pave the way for further studies to clarify the potential role of NKB in placental abnormalities characterizing PCOS. The manuscript is well written. This work, which addresses a current issue, could be published after some improvements to the manuscript.

Page 2, line 93. “from the Department of Obstetrics and Gynecology of the University Hospital of Patras”: specify the country (Greece).

Page 3, lines 116-177. “according to the manufacturer’s protocol (including a 15-minute DNAse I treatment).”: this sentence is not sufficient for a scientific article: please briefly describe the technique used.

Page 3, lines 121-122. “according to the manufacturer’s instructions.” : Likewise, please briefly describe the technique used.

Page 3, lines 133-134. “Samples were immediately centrifuged,”: What were the characteristics of centrifugation? (Speed in g number, duration).

Page 3, lines 136-137. “were measured by electrochemiluminescence quantization in the Cobas e601 analyzer”: Briefly describe the technique used.

Page 5, table 3: use italics to write the names of genes.

Author Response

REVIEWER 1

“Polycystic ovary syndrome could occur in utero, and involve the placenta in particular through the production of kisspeptin and neurokinin B. The study presented here concerns a comparison of the expression of placental KISS1, NKB and neurokinin receptors in the placental mRNA of patients with polycystic ovarian syndrome and of healthy pregnant women, with the aim of correlating these gene expressions with maternal and cord blood sex steroid levels. The findings pave the way for further studies to clarify the potential role of NKB in placental abnormalities characterizing PCOS. The manuscript is well written. This work, which addresses a current issue, could be published after some improvements to the manuscript”.

Thank you very much for your kind comments.

“Page 2, line 93. “from the Department of Obstetrics and Gynecology of the University Hospital of Patras”: specify the country (Greece)”.

Done, thank you (see line 95).

“Page 3, lines 116-177. “according to the manufacturer’s protocol (including a 15-minute DNAse I treatment).”: this sentence is not sufficient for a scientific article: please briefly describe the technique used”.

Done (see lines 118-123), thank you.

“Page 3, lines 121-122. “according to the manufacturer’s instructions.” : Likewise, please briefly describe the technique used”.

Done (see lines 124-128), thank you.

“Page 3, lines 133-134. “Samples were immediately centrifuged,”: What were the characteristics of centrifugation? (Speed in g number, duration)”.

We have added this information (see line 140), thank you.

“Page 3, lines 136-137. “were measured by electrochemiluminescence quantization in the Cobas e601 analyzer”: Briefly describe the technique used”.

Done (see lines 144-153), thank you.

“Page 5, table 3: use italics to write the names of genes”.

Done, thank you.

Reviewer 2 Report

Comments and Suggestions for Authors

The study attempted to tackle an important and interesting topic. I did not find any major issue associated with the design of the study or experimental procedures. However, some minor points should be addressed before considering the paper for publication:

1.      Abstract, lines 22-23: please, add “regardless of PCOS” to “NKB expression depended on fetal gender, being higher in pregnancies with male fetuses”.

2.      Figure 1: Since the data were analyzed by Mann-Whitney test, they should be presented as boxplots with interquartile intervals (minimum and maximum values). The individual values shown as dots will be helpful as well.

3.      Tables: please indicate the meaning of values with brackets in the comments.

4.      The authors calculated a lot of correlations between hormones, but the analysis of these data is insufficient. What is physiological meaning of such correlations? Besides, the discussion is mostly focused on the cited literature data and poorly analyses the results obtained in this particular study, especially because the authors found very interesting phenomenon associated with increase of NKB in the pregnancies with female newborns, which can be linked with inherited nature of PCOS. Next, a great difference in NKB, NKRs and KISS1 mRNA expression between male and female pregnancies should be explained.

Author Response

The study attempted to tackle an important and interesting topic. I did not find any major issue associated with the design of the study or experimental procedures. However, some minor points should be addressed before considering the paper for publication:

  1. Abstract, lines 22-23: please, add “regardless of PCOS” to “NKB expression depended on fetal gender, being higher in pregnancies with male fetuses”.

Done, thank you (see lines 23-24).

  1. Figure 1: Since the data were analyzed by Mann-Whitney test, they should be presented as boxplots with interquartile intervals (minimum and maximum values). The individual values shown as dots will be helpful as well.

Thank you for your comment. In Figure 1, we present the placental expression of NKB in PCOS relative to controls. The median expression of NKB (using ACTB as the housekeeping gene) in controls is considered as reference and therefore has a value of 1, and the value for PCOS is the ratio of the median expression of NKB in PCOS / median expression of NKB in controls. For example, in the female offspring group, the median expression of NKB in PCOS and Controls is 0.0008 and 0.0001, respectively; therefore, the values in the figure are 1 for Controls (arbitrary) and 8 for PCOS (= 0.0008 / 0.0001). The Mann-Whitney test was used to compare the absolute values of the placental expression of NKB (assessed as relative expression to ACTB by the ΔΔCT method) between the two groups. We chose to produce a figure with the relative (PCOS to Controls) rather than absolute values of NKB expression, because we think it better presents our findings and makes it easier for the reader to understand the magnitude of the difference between the groups. Besides, it is common in the literature to present data regarding comparison of gene expression this way. However, if you think it is necessary, we could replace the figure or add a figure showing the absolute data.

  1. Tables: please indicate the meaning of values with brackets in the comments.

Done, thank you.

  1. The authors calculated a lot of correlations between hormones, but the analysis of these data is insufficient. What is physiological meaning of such correlations? Besides, the discussion is mostly focused on the cited literature data and poorly analyses the results obtained in this particular study, especially because the authors found very interesting phenomenon associated with increase of NKB in the pregnancies with female newborns, which can be linked with inherited nature of PCOS. Next, a great difference in NKB, NKRs and KISS1 mRNA expression between male and female pregnancies should be explained.

Thank you for your comments. We have made additions in the discussion section, trying to provide plausible explanations for our findings (see lines 313-323, 324-328, 404-406, 411-418, and 424-428).  

Round 2

Reviewer 2 Report

Comments and Suggestions for Authors

None